# Active thermokarst regions contain rich sources of ice nucleating particles

Kevin R. Barry[1]*, Thomas C. J. Hill[1], Marina Nieto-Caballero[1], Thomas A. Douglas[2], Sonia M. Kreidenweis[1], Paul J. DeMott[1], and Jessie M. Creamean[1]

[1]Department of Atmospheric Science, Colorado State University, 1371 Campus Delivery, Fort Collins, Colorado 80523-1371, United States of America
[2]U.S. Army Cold Regions Research and Engineering Laboratory, 9th Avenue, Building 4070, Fort Wainwright, AK, USA 99703

*Correspondence to*: Kevin R. Barry (Kevin.Barry@colostate.edu)

**Abstract.** Rapid Arctic climate warming, amplified relative to lower latitude regions, has led to permafrost thaw and associated thermokarst processes. Recent work has shown permafrost is a rich source of ice nucleating particles (INPs) that can initiate ice formation in supercooled liquid clouds. Since the phase of Arctic clouds strongly affects the surface energy budget, especially over ice-laden surfaces, characterizing INP sources in this region is critical. For the first time, we provide a large-scale survey of potential INP sources in tundra terrain where thermokarst processes are active and relate to INPs in the air. Permafrost, seasonally thawed active layer, ice wedge, vegetation, water, and aerosol samples were collected near Utqiaġvik, Alaska in late summer and analyzed for their INP contents. Permafrost was confirmed as a rich source of INPs that was enhanced near the coast. Sensitivity to heating revealed differences in INPs from similar sources, such as the permafrost and active layer. Water, vegetation, and ice wedge INPs had the highest heat labile percentage. The aerosol likely contained a mixture of known and unsurveyed INP types that were inferred as biological. Arctic water bodies were shown to be potential important links of sources to the atmosphere in thermokarst regions. Therefore, a positive relationship found with total organic carbon considering all water bodies gives a mechanism for future parameterization as permafrost continues to thaw and drive regional landscape shifts.

## 1 Introduction

The Arctic landscape is sensitive, dynamic, and changing, with many of the shifts connected to the permafrost. Permafrost, earth material like soil, ice, rocks, and organic matter that remains frozen for more than two years, underlies approximately 22% of the Northern Hemisphere landmass (Obu et al., 2019), and is rapidly thawing from Interior Alaska to the Arctic (Douglas et al., 2021; Farquharson et al., 2022; Streletskiy et al., 2015; Streletskiy et al., 2017). Melting of ice-rich

permafrost leads to development of "thermokarst", which includes the formation of water bodies called thermokarst lakes (TKLs). The formation and drainage of TKLs strongly impact surrounding ecosystems, and additionally TKLs are sources of methane and carbon dioxide to the atmosphere (Jorgenson, 2013; Walter et al., 2006). Other thermokarst landforms include retrogressive thaw slumps, slope failures often triggered by the flow of material from the top seasonally frozen and thawed active layer, and thermokarst troughs and pits, low-lying areas created when ice-rich permafrost or massive ice features like ice wedges degrade. It was estimated in a study in Prudhoe Bay, Alaska, that 23% of the surface ice wedges degraded between 1949 and 2012, leading to major impacts on the composition of the vegetation (Jorgenson et al., 2015). Permafrost coasts across the Arctic are increasingly sensitive to erosion, the loss of which has environmental and economic impacts (Irrgang et al., 2022).

In addition to landforms created, thawing permafrost has broad atmospheric impacts, as it can potentially alter clouds by serving as a source of ice nucleating particles (INPs) (Barry et al., 2023; Creamean et al., 2020). INPs are particles that trigger ice formation in clouds, and are necessary to initiate ice formation warmer than -38 °C (level of homogeneous freezing). They can alter the surface energy budget by impacting the cloud phase and optical thickness, as Arctic liquid clouds strongly contribute to a positive cloud forcing (Shupe & Intrieri, 2004). Replacement of ice with liquid in clouds has been shown to strengthen Arctic amplification, which is the enhanced regional warming due to phenomena such as the ice-albedo feedback (Tan & Storelvmo, 2019). Sources of INPs include: biological material such as proteins from certain species of bacteria and fungi active at temperatures up to and warmer than -5 °C; mineral dust that is efficient below about -15 °C; and complex organics that are effective over the entire temperature range (e.g. Hill et al., 2018; Murray et al., 2012; Testa et al., 2021; Tobo et al., 2014). Thawed permafrost material was shown to have comparable ice nucleation activity to midlatitude and glacial soil dust (Creamean et al., 2020). If the material enters TKLs, its persistence in the water and release in lake spray aerosol could persist for weeks, with ice nucleation activity on a surface area basis up to and exceeding that of mineral dust (Barry et al., 2023). Moreover, the majority of the INPs were inferred to be of biological and organic origin and highly active at relatively warm temperatures (Creamean et al., 2020), and therefore could impact the lifetime of long-lasting Arctic mixed-phase clouds that commonly exist between -25 and -5 °C (Morrison et al., 2012). Although permafrost is a massive reservoir of INPs, it is not represented as a source in global or regional climate models. Models struggle to accurately represent Arctic clouds, with current ice microphysical parameterizations thought to be a large contributor to biases (Taylor et al., 2019), underscoring the value of Arctic INP measurements.

Previous Arctic INP measurements have largely focused on collecting air samples from ships, aircraft, or fixed ground-based sites (e.g., Bigg, 1996; Hartmann et al., 2021; Mason et al., 2016; Prenni et al., 2007; Wex et al., 2019). Recent studies have noted evidence of increased INP concentrations in terrestrial airmasses (Conen et al., 2016; Creamean et al., 2018; Irish et al., 2019; Šantl-Temkiv et al., 2019). Most recently, a year-long observation of INPs in the central Arctic revealed a seasonal dependence with the highest concentrations found in summer (Creamean et al., 2022a), similar to trends observed in other Arctic work (e.g., Creamean et al., 2018; Wex et al., 2019). Despite several Arctic studies, a comprehensive source-based analysis has not been done. In the ARCtic Study of Permafrost Ice Nucleation (ARCSPIN) of September 2021, we

surveyed several previously-uncharacterized potential sources of airborne terrestrial-based Arctic INPs in a region underlain by continuous permafrost. Permafrost and ice wedge cores, active layer, vegetation, sediment, and water samples were collected at peak thaw in late summer to profile their INP contents and relate to coincident air measurements.

## 2 Methods

### 2.1 Measurement overview

The ARCSPIN sampling campaign was conducted from September 1-17, 2021, within and near Utqiaġvik, Alaska. Its surficial geology is categorized as marine silt and sand, where permafrost temperatures have increased by 0.85 °C (-8.532 to -7.678 °C) at 20 m between 2009 and 2021 (Romanovsky, 2021). The lowland landscape is dominated by patterned ground comprised of ice wedge polygons that are actively undergoing thermokarst processes (Farquharson et al., 2016). Common vegetation in this region includes sedge, grass, moss, rush, dwarf-shrub, and forb (Raynolds et al., 2006).

The overview of all sampling days is detailed in Table 1 and Figure 1. Half (6) of the days focused on downwind TKL (both fresh and brackish) measurements, where, if feasible, upwind measurements were included (3 of 6 days). Upwind and downwind locations were determined by the wave movement and wind direction. All wind data came from the Wiley Post–Will Rogers Memorial Airport weather station (PABR). Other periods focused on sampling in the saline lagoon with a small boat (3 days) and coastal estuarine and oceanic sampling (3 days). Sites were chosen based on accessibility with all-terrain vehicles (ATVs) as well as to maximize areal coverage and diversity of terrain and weather conditions (e.g., targeting onshore versus offshore winds). Additionally, pre-campaign water measurements were made at one location in the Chukchi Sea (71.32921429 °N, 156.678083 °W), approximately 2 meters from the coast, on August 22-24 to sample conditions during (22[nd]) and post (23[rd] and 24[th]) stormy weather. The storm had minimal precipitation, and was instead marked by strong winds and waves, with average sustained winds of 7.6 (gust>13), 3.1, and 1.8 m s$^{-1}$, respectively, on the 3 days.

At each measurement site, coastal and lake-shore aerosol filters, TKL or ocean water, sediment, permafrost, ice wedge, active layer, and vegetation samples were collected. Aerosol for INP analyses was collected onto 0.2 µm Nuclepore track-etched membranes (Whatman) in disposable filter units (Nalgene) with a battery-powered pump (Gilian 12). The filters were precleaned before loading by brief ultrasonication (2 ×10 s) in methanol followed by two 0.1 µm filtered deionized (DI) water rinses (Barry et al., 2021). The sampling height was approximately 1.5 m, and filters were collected after sampling 2 to 4 hours, depending on site.Typical flow rates were 7 standard L (sL: 0 °C, 1013.25 mb) min$^{-1}$, and the average total volume of air filtered per sample was 1350 sL. The filter setup in the field locations is shown in Figure 2. Additionally, filters were collected at the U.S. Department of Energy Atmospheric Research Measurment North Slope of Alaska (DOE ARM NSA; herewithin: DOE) facility (Fig. 1). Five samples were collected between 2 and 22 hours, with sample length determined by the consistency of the wind direction. The wind directions covered were out of the S, SE, E, NE, and NW. The average flow rate was 22 sL min$^{-1}$, which resulted in an average volume of air filtered of 17400 sL at a sampling height of approximately 10 m. Additional aerosol samples were collected for DNA analyses, but are not presented in this work.

Water samples were collected into a prerinsed (with sample) 500 mL bottle (Nalgene), before placed into sterile 15 or 50 mL tubes (Corning). Water was collected at the surface and near the bottom of the TKLs (depth 0.6-2 m) and coastal ocean (depth ~1.5 m) with a kayak and horizontal water sampler (Pentair), up to 70 m from the shoreline. TKL and oceanic

sediment samples were collected with a Universal Corer (Aquatic Research Instruments) approximately 5-10 cm below the floor from the same location as the water sample, and subsamples were placed into 1-oz Whirl-Pak bags. Permafrost and ice wedge cores in proximity to the water were taken with an 8 cm diameter Snow, Ice, and Permafrost Research Establishment (SIPRE) auger, and 2-4 subsamples were taken at various depths along the core (based on visual differences in composition) and packed into Whirlpak bags. The average core length was 84 cm. A corresponding active layer sample was taken directly

above each permafrost core and placed into 1-oz Whirl-Pak bags. Representative vegetation clippings were collected into plastic slider bags, weighed, 250 mL of DI water added, shaken, and poured into a sterile 15 mL tube (Corning). All samples were stored in a cooler at the measurement site, and then in a -20 °C freezer in Utqiaġvik at the Naval Arctic Research Laboratory for the duration of the campaign. They were subsequently transported frozen in coolers back to Colorado State University (CSU) and stored at -20 °C until analysis.

**2.2 Sample analysis**

Samples were analyzed for INP concentrations, each as a function of temperature, with the CSU Ice Spectrometer (IS; Creamean et al., 2022b; DeMott et al., 2018). Sediment, active layer, ice wedge, and permafrost samples were thawed, stirred, and a suspension made by weighing approximately 2 g of material and combining it with 20 mL of DI water. Filters were resuspended in 7-8 mL of 0.1 µm filtered DI water. Due to the abundance of INPs, dilution series were made with

suspensions and water samples in 0.1-µm-filtered DI water: 11-fold dilutions for the aerosol (400 µL sample and 4000 µL 0.1-µm-filtered DI water) and 20-fold dilutions (250 µL sample and 4750 µL 0.1-µm-filtered DI water) for all other samples. Suspensions and their corresponding dilutions were dispensed in blocks of 32, 50 µL aliquots in single-use 96-well PCR trays (Optimum Ultra), along with a 32-well negative control of 0.1-µm-filtered DI water. The trays were placed into the aluminum blocks of the IS, cooled at a rate of 0.33 °C min$^{-1}$, freezing detected optically with a CCD camera, and 1-s data recorded. Next,

the frozen fractions were converted to cumulative INP concentrations per mL of water, per L of air (considering the volume of air filtered and resuspension volume), or per g of material (considering the weight of material and resuspension volume) (Vali, 1971). 95% confidence intervals were computed following Agresti and Coull (1998). In total, 20 aerosol, 47 water, 20 permafrost, 8 sediment, 11 ice wedge, 6 active layer, and 5 vegetation washing samples were processed.

Aerosol INP concentrations were corrected from the average of two blanks that were prepared, transported, and

processed identically, except that no airflow was sent through them, by subtraction of the average INPs per filter as a function of temperature before conversion to concentration. These corrections were minor since, for example, there were only an average of 73 INPs per blank filter at -28 °C where there were typically around 5000 INPs at -28 °C even in many of the lower volume tundra air samples. Undiluted estuarine and seawater samples were corrected for freezing point depression (FPD), based upon measured conductivity in the field (Extech EC400) and at CSU for the saltier samples (Extech EC150). Samples

were normalized to the average measured conductivity of seawater samples of 51383 µS cm$^{-1}$ corresponding to a 1.8 °C FPD,

resulting in a lagoon correction of 1.2 °C and brackish TKL correction of 1.1 °C. The dilutions were not adjusted since they were prepared with 0.1-µm-filtered DI water. Thermal treatments were done on select samples for insight into sample composition. 2.4 mL of selected samples were heated at 95 °C for 20 min and retested on the IS to determine the heat labile fraction of INPs. This treatment has been used extensively in the past on samples from diverse environments (e.g. Kanji et al.,

2017; McCluskey et al., 2018; Suski et al., 2018), to estimate contributions of INPs that are inferred to be of proteinaceous origin.

Total organic carbon (TOC) concentrations were measured in a subset of representative water samples (28 of 47 total) by injecting 3 mL of sample into a TOC-VCSH (Shimadzu). Total carbon was first determined through combustion at 680 °C, creating $CO_2$, and inorganic carbon was determined through sample acidification followed by sparging to additionally create

$CO_2$. The $CO_2$ was detected and compared to calibration curves. TOC was calculated by subtracting the inorganic carbon from the total carbon. The background TOC was subtracted by injecting 3 mL of DI water in sample tubes 3 times and taking the average.

Principal component analysis (PCA) was performed on all INP-temperature spectra over the temperature interval from -6 to -20 °C. This range was chosen because the majority of data had measurements and definable characteristics over

this interval (as many spectra are similarly log-linear, at colder temperatures). Samples that did not have complete measurements in this range were either interpolated or extrapolated. To build the matrix for analysis, the slope in 2 degree temperature intervals was calculated (change in $log_{10}$[INP concentration]/change in temperature). Another potential defining variable, $log_{10}$ of the ratio of the average INP concentration in each 2 degree temperature interval and the average INP concentration at -12 °C, was also calculated. In total, there were 121 samples included, each with 14 variables. Next, the

sampling dimension mean was removed, all variables were standardized, and the temporal covariance matrix calculated before performing eigenanalysis.

| Date (2021) | Name | Latitude (°) | Longitude (°) | Environment/ Collection type | Samples analyzed |
|---|---|---|---|---|---|
| 1-Sep | Emaiksoun Lake | 71.25057 | -156.77317 | Thermokarst lake (TKL) | DA, P, S, TW |
| 2-Sep | Untitled Lake 1 | 71.23529 | -156.30406 | TKL: upwind and downwind | DA, UA, AL, P, I, S, TW, V |
| 4-Sep | DOE | 71.32272 | -156.61506 | Fixed | A |
| 5-Sep | Point Barrow | 71.38535 | -156.46100 | Ocean and lagoon | A, LW, SW |
| 6-Sep | Nunavak Bay | 71.25240 | -156.87332 | TKL (brackish) | DA, AL, P, I, S, TW, V |
| 7-Sep | Elson Lagoon | 71.29581 | -156.26890 | Lagoon | A, LW |
| 8-Sep | Will Rogers/ Wiley Post Monument | 71.15311 | -157.06609 | Ocean: onshore | A, AL, P, S, TW, V |
| 9-Sep | Elson Lagoon DOE | 71.25556 71.32272 | -156.01580 -156.61506 | Lagoon Fixed | A, LW, SW A |
| 11-Sep | Untitled Lake 2 | 71.23806 | -156.60472 | TKL: upwind and downwind | DA, UA, AL, P, S, TW, V |
| 12-Sep | DOE | 71.32272 | -156.61506 | Fixed | A |
| 13-Sep | Mayoeak River | 71.25915 | -156.44528 | TKL (brackish) | DA, P, I, S, TW, V |
| 14-Sep | Will Rogers/ Wiley Post Monument | 71.15312 | -157.06609 | Ocean: offshore | A, AL, P, I, S |
| 15-Sep | Elson Lagoon DOE | 71.31522 71.32272 | -156.29845 -156.61506 | Lagoon Fixed | A, LW, SW A |
| 17-Sep | Emaiksoun Lake | 71.23097 | -156.77237 | TKL: upwind and downwind | DA, UA, AL, P, I, S, TW, V |
| 18-Sep | DOE | 71.32272 | -156.61506 | Fixed | A |

**Table 1:** Name, latitude, longitude, environmental location, and list of processed sample types for the main sample collections. Lagoon locations on September 7, 9, and 15 refer to an average latitude and longitude of the collected samples, and the locations of TKLs with both upwind and downwind collections refer to the downwind measurement site. "DOE" refers to the location of the fixed Department of Energy site. For "Samples analyzed", DA=Downwind Aerosol, UA=Upwind Aerosol, A=Aerosol, AL=Active Layer, P=Permafrost, I=Ice Wedge, S=Sediment, LW=Lagoon Water, SW=Seawater, TW=Thermokarst Lake Water, V=Vegetation.

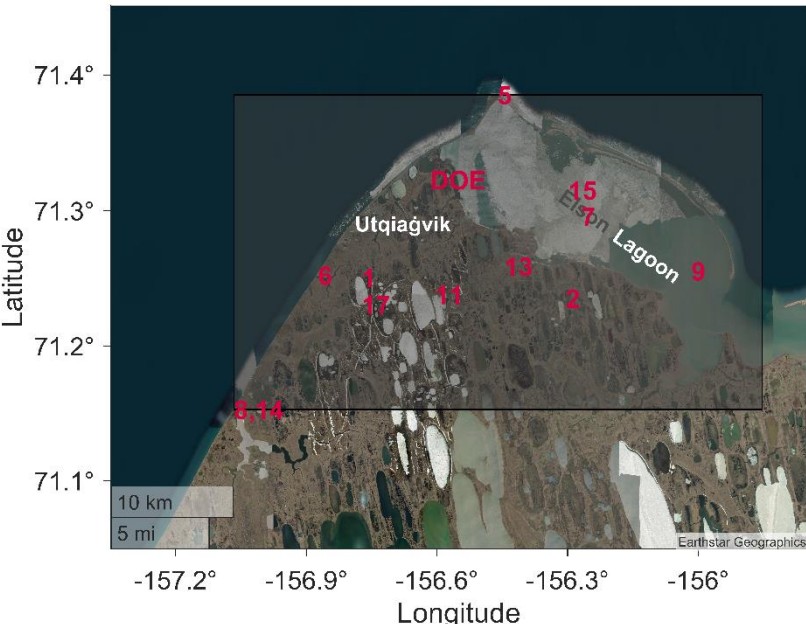

**Figure 1:** Area map showing the bounding box of all samples obtained in black, and the location of the specific sampling days (in September 2021) from Table A1 in red. "DOE" refers to the location of the fixed site. The locations of TKLs with both upwind and downwind collections refer to the downwind measurement site.

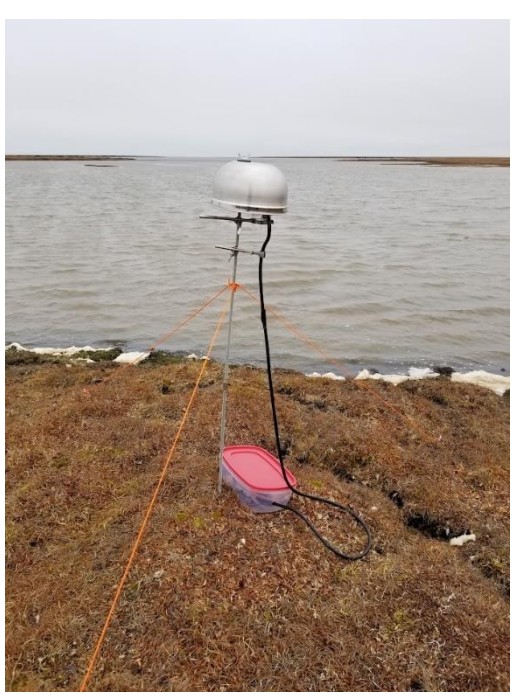

**Figure 2:** Sampling setup for measuring ice nucleating particles (INPs) in the aerosol near a thermokarst lake.

## 3 Results and Discussion

### 3.1 Overview of Arctic INP measurements

INP concentrations active at -15 °C measured in the ambient aerosol during ARCSPIN ranged from 0.0009 to 0.4 L$^{-1}$ (average 0.04 L$^{-1}$) (Fig. 3; spectra in Fig. S1). -15 °C is focused upon for comparison to previous Arctic INP measurements and due to its relevance for Arctic mixed-phase clouds (Morrison et al., 2012). The highest values were found downwind of TKLs, near the coast, and over Elson Lagoon (Figs. 3-4). Among aerosol filter sample types (TKL, ocean, lagoon, and DOE), the DOE had the lowest variability (SD: 0.0076 L$^{-1}$) while the lagoon had the highest variability (SD: 0.23 L$^{-1}$), attributed to sampling diverse sources from a moving boat.

To test the contribution of water bodies to atmospheric INPs, a pairwise t-test indicated increased INP concentrations downwind of TKLs at 95% confidence in all three cases at -18 to -23 °C and at the coldest temperatures (-23 to -28 °C) for September 17$^{th}$ only (Fig. S2). Wind speeds were variable, with the averages on the 2$^{nd}$, 11$^{th}$, and 17$^{th}$, of 9.3, 2.9, and 6.2 m s$^{-1}$ respectively, near and above the 3.5 m s$^{-1}$ threshold found by Slade et al. (2010) for freshwater aerosol enhancement. There was no obvious correlation between INP concentrations and wind speed, which agrees with recent relative insensitivity found between freshwater aerosol mass flux and wind speed (Harb & Foroutan, 2022). We conclude that TKLs can generate INPs, but their impact on ice nucleation activity may be freezing temperature-dependent. For the potential of coastal INP enhancement, a comparison between wind directions at one location was made on September 8$^{th}$ and 14$^{th}$ (Table 1 and Fig. 1; values in Fig. 3). INP concentrations were higher across all measured temperatures at 95% confidence when the wind was onshore (average direction on the 8$^{th}$ of 246°) from the Chukchi Sea compared with offshore (average on the 14$^{th}$ of 115°). This is despite slightly greater average wind speeds on the 14$^{th}$ (5.1 vs. 4.3 m s$^{-1}$).

This analysis provides evidence for water bodies as potential vessels for transporting INPs to the air under wind stress or alternative mechanisms such as methane bubbling up through TKLs, called ebullition (Walter et al., 2006). Primary marine aerosol has been found to have a power law relationship with wind speed over the Southern Ocean (Moore et al., 2022; Sanchez et al., 2021) and North Atlantic (Saliba et al., 2019), and an exponential relationship was found between particle concentration and wind speed over the Arctic Ocean (Leck et al., 2002). Enhancement in aerosol does not necessarily translate to enhancement of INPs. Further, the wind speeds experienced during ARCSPIN were typically less than 10 m s$^{-1}$, with an average of 6 m s$^{-1}$ for the month of September. At the DOE, which was removed from local direct inputs, there was consistency in INP concentrations regardless of wind direction (captured from terrestrial and marine sources) and average wind speed (ranging from 3.9-7.2 m s$^{-1}$), suggesting a background level of INPs in the air coexisting with periodic coastal and TKL enhancement.

The aerosol INPs measured during ARCSPIN varied in comparability to other terrestrial-based Arctic campaigns, partially attributed to seasonality. The concentrations were higher than Creamean et al. (2018), who measured an average INP concentration of 0.005 L$^{-1}$ at -15 °C between March and May at Oliktok Point, and Mason et al. (2016), who also found an average INP concentration of 0.005 L$^{-1}$ at -15 °C at Alert between March and July. The INPs were more similar to those

reported by Šantl-Temkiv et al., (2019), who measured an average INP concentration of 0.07 L⁻¹ at -15 °C in August at Villum Research Station. Wex et al. (2019), who measured INPs at Utqiaġvik for a year, found concentrations up to 0.01 L⁻¹ at -10 °C during September, which is within the range and time period of ARCSPIN (Fig. S1).

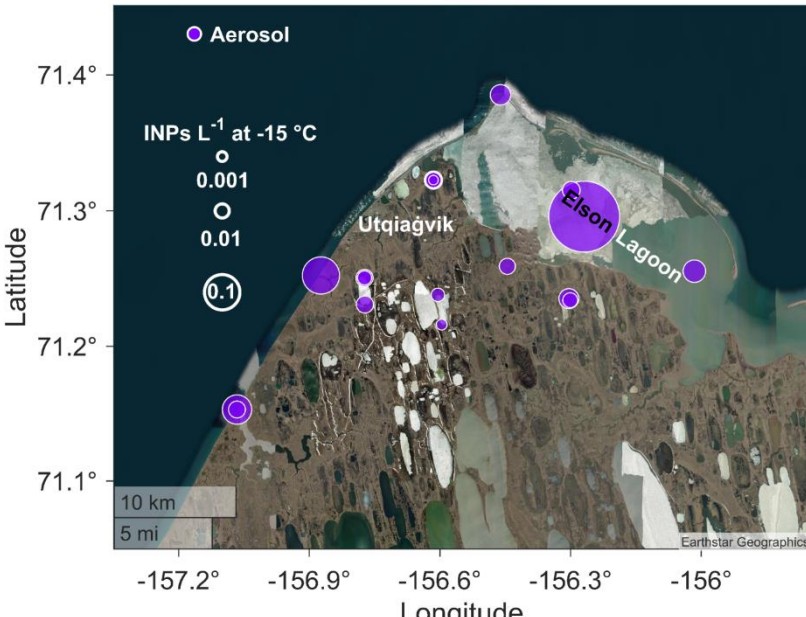

**Figure 3:** INP concentration per L air at -15 °C for aerosol samples (purple). The size of the markers corresponds to the INP concentration. Lower left coastal samples were sampled on September 8th and September 14th.


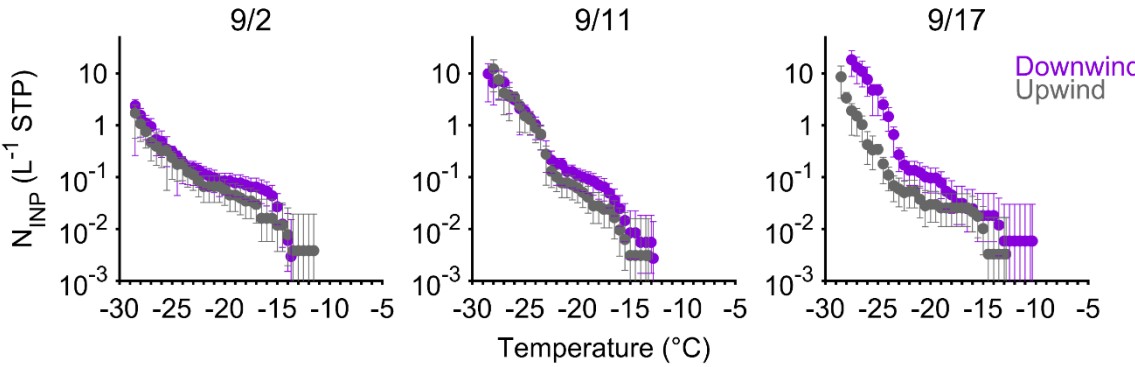

**Figure 4:** Cumulative INP-temperature spectra for the three cases that sampled aerosol upwind (gray) and downwind (purple) of a thermokarst lake. 95% confidence intervals are plotted (any confidence intervals overlapping with 0 are not shown).

In addition to TKLs serving as sources for aerosolized INPs, INP concentrations in water samples were highest (but most variable) in the TKLs, followed by the lagoon and seawater, respectively (Fig. 5; spectra in Fig. S3). At -15 °C, the average TKL INP concentration was 120,000 mL$^{-1}$ (SD=178,000 mL$^{-1}$) compared with 31,000 mL$^{-1}$ (SD=17,000 mL$^{-1}$) in the lagoon and 17,000 mL$^{-1}$ (SD=19,000 mL$^{-1}$) in seawater. However, stormy conditions increased INPs in seawater from 1,300 mL$^{-1}$ to 63,000 mL$^{-1}$ at -15 °C. Previous measurements from the Bering Strait and Chukchi Sea (Creamean et al., 2019)

measured INP concentrations of 100-3000 mL$^{-1}$ at -15 °C, much lower than ARCSPIN. Other ship-based Arctic measurements farther from land (Hartmann et al., 2021; Wilson et al., 2015) reported INP concentrations less than 100 mL$^{-1}$ at -15 °C in bulk seawater during summer. Therefore, the weather conditions, type of water body, and proximity to the coast are all important for determining water INP concentrations.

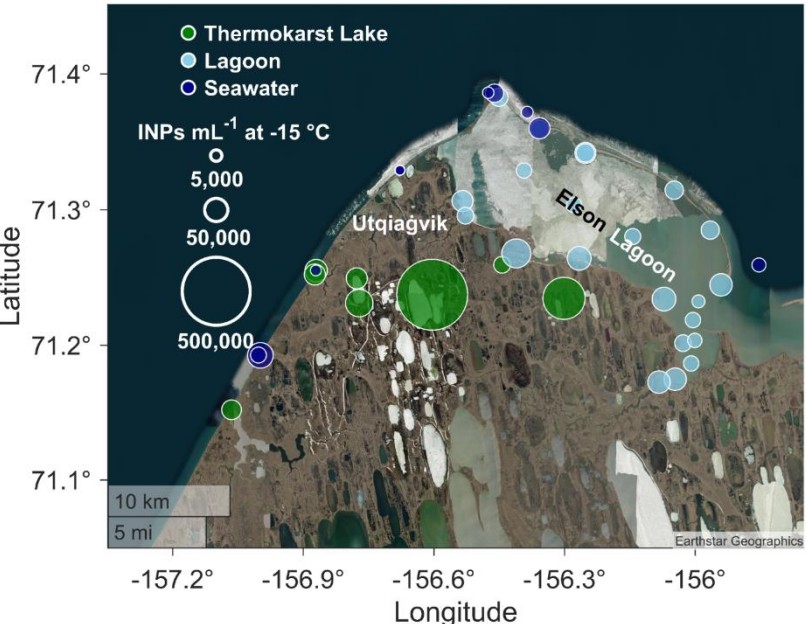

**Figure 5:** INP concentration per mL water at -15 °C for thermokarst lake (TKL) (green), lagoon water (light blue), and seawater (dark blue) samples. The size of the markers corresponds to the INP concentration.

        Next, we measured many of the terrestrial-based sources in a thermokarst region that may contribute to the Arctic INP budget. Soil and vegetation samples were rich sources of INPs (Fig. 6; spectra in Fig. S4), with permafrost having up to

$5\times10^8$ INPs g$^{-1}$ at -15 °C (average of $1.1\times10^8$ g$^{-1}$, SD=$1.4\times10^8$ g$^{-1}$). The highest permafrost values were found near the coast, suggesting a prodigious reservoir of INPs that could be released into water bodies undergoing coastal erosion or thermokarst degradation. Permafrost sampled at different core depths showed similar INP concentrations (Fig. S5) and, therefore, only the sample closest to the surface is presented. The permafrost INP spectra largely agree in concentration with permafrost samples from Fairbanks, Alaska, in Creamean et al. (2020) and with glacial outwash sediments from Svalbard, Norway (Tobo et al.,

2019), with values between $10^8$ and $10^9$ g$^{-1}$ at -15 °C. The comparability to ARCSPIN, despite differences in collection depths and locations, is promising for modelling of permafrost sources.

Lake and ocean sediment contained up to $10^8$ INPs g$^{-1}$ (average of 3.2x10$^7$ g$^{-1}$, SD=3.5x10$^7$ g$^{-1}$) at -15 °C, with the highest values in sediment found inland within freshwater TKLs and lower values found near and from the Chukchi Sea. Despite the ocean sediment being a lower source of INPs, its suspension would have contributed to the 50-fold INP increase

observed in Chukchi Sea water during the stormy period. Vegetation washings contained lower levels of INPs overall (average of 2x10$^6$ g$^{-1}$, SD=1.7x10$^6$ g$^{-1}$ at -15 °C), but were the source of the warmest temperature INPs (Fig. S4), with a detected freezing onset as warm as -4 °C, likely indicative of populations of ice nucleation-active bacteria (e.g., Hill et al., 2014; Huang et al., 2021). Active layer samples (Figs. S6 and S7) had similar ice nucleation activities to permafrost, with an average concentration of 3.6x10$^8$ g$^{-1}$ at -15 °C. Ice wedges (Figs. S6 and S7) had values comparable to TKLs (assuming a density of 1 g mL$^{-1}$), with

an average of 1.7x10$^5$ g$^{-1}$ (SD=1.7x10$^5$ g$^{-1}$) at -15 °C. The similarity between ice wedges and TKLs is ascribed to snow melt being the dominant source of ice wedge ice and contributing to a majority of TKL water. Thermokarst landscapes are therefore comprised of several potent (and deep) INP reservoirs that are comparable with midlatitude soils and could influence Arctic clouds.

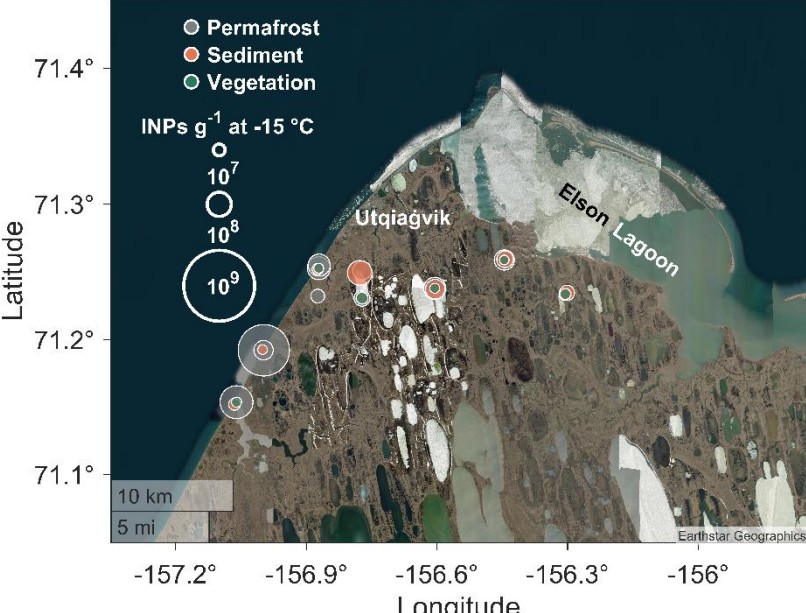

**Figure 6:** INP concentration per g at -15 °C for permafrost (gray), lake and ocean sediment (salmon), and vegetation (dark green) samples. The size of the markers corresponds to the INP concentration.

**3.2 Source separation and characterization**

To better understand similarities and differences among thermokarst landscape sources and aerosol samples, we next present PCA and heat treatment results. PCA was applied to visualize how samples clustered within and between categories to relate INPs from sources to the aerosol. Figure 7 shows all source and aerosol samples colored by sample type. PC1 explains 47% of the variance while PC2 explains an additional 23% of the variance. Among sources, PC1 and PC2 separate water and vegetation (negative PC1 values) from permafrost and sediment samples (positive PC1 and negative PC2 values). The aerosol spans the range of collected samples along PC1, but is separated on PC2.

Samples can be further analyzed for characteristic INP differences through response to heating. In Figure 8, suspensions of all samples for September 2$^{nd}$ and 17$^{th}$ were heated to 95 °C to divide the INPs into heat labile and stable fractions. The heat labile fraction identifies putative biological INPs through protein denaturation, and the heat stable fraction may be organic or mineral. Among sources, the TKL water, ice wedge, and vegetation had the highest fractions of heat labile INPs, with above 95% at -10 °C, and above 90% at -15 °C except for the ice wedge on the 2$^{nd}$ (Fig. 8). The permafrost, active layer, and TKL sediment samples had high fractions of heat labile INPs at -10 °C (>70%: except for the sediment on the 2$^{nd}$), while only the active layers and sediment sample from the 17$^{th}$ had greater than 50% heat labile INPs at both -15 and -20 °C.

The PCA revealed that while most sources, especially water and vegetation washings, cluster together within their groups, indicating relative homogeneity across location and time, there was substantial variability among permafrost. This suggests permafrost INP composition may be more heterogeneous, despite concentration comparibility to previous work presented in section 3.1. Both methods reveal differences between the permafrost and active layer, which were similar in INP concentrations (Table S1), but have relative PCA separation (Fig. 7) and heat sensitivity (Fig. 8), suggesting dissimilar INP populations. TKL sediments may also harbor distinct INP populations from other samples, with some uniqueness in heat labile fractions (Fig. 8), but similar values to permafrost on the PCA (Fig. 7). TKL sediments are comprised of former permafrost, but undergo loss of organic carbon, nitrogen, and phosphorous during the transition (Ren et al., 2022), which could have contributed to both measured similarities and differences to permafrost seen with the PCA and heat treatment. These tools should be seen as complimentary in providing pieces of evidence, as PCA utilizes the spectral shape of the original sample, while the heat treatment compares proportions in the two spectra.

All aerosol samples contained abundant heat labile INPs, with fractions of nearly 100% at all temperatures (Fig. 8), With PCA, the aerosol spans the range of collected samples along PC1, suggesting diverse sampled source types contributing to the collected airborne INPs (Fig. 7). As many source samples contained abundant heat labile INPs, with the most prominent being the TKL water, ice wedge, and vegetation, this analysis agrees with the PCA that multiple sources could have contributed. However, the PCA also uncovers that the aerosol are separated from most other samples along PC2, implying the aerosol additionally contained unsurveyed sources (e.g. other local or long range transport).

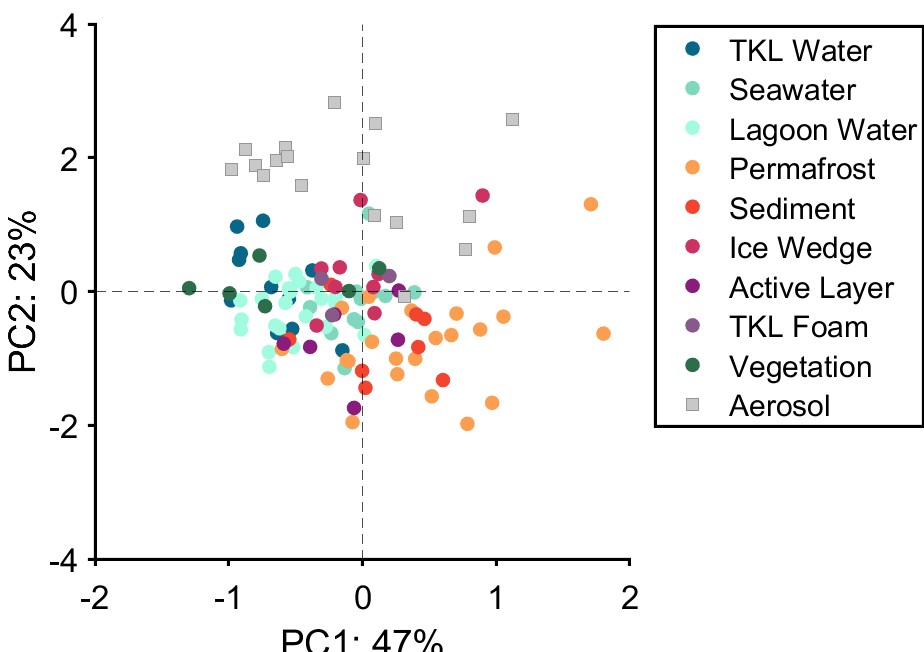

**Figure 7:** Principal component analysis for all processed samples, broken down by sample type, based upon sample slope and a midpoint concentration ratio between -6 and -20 °C. Percent variance explained is given on each respective axis. The axes limits are scaled by the variance explained.



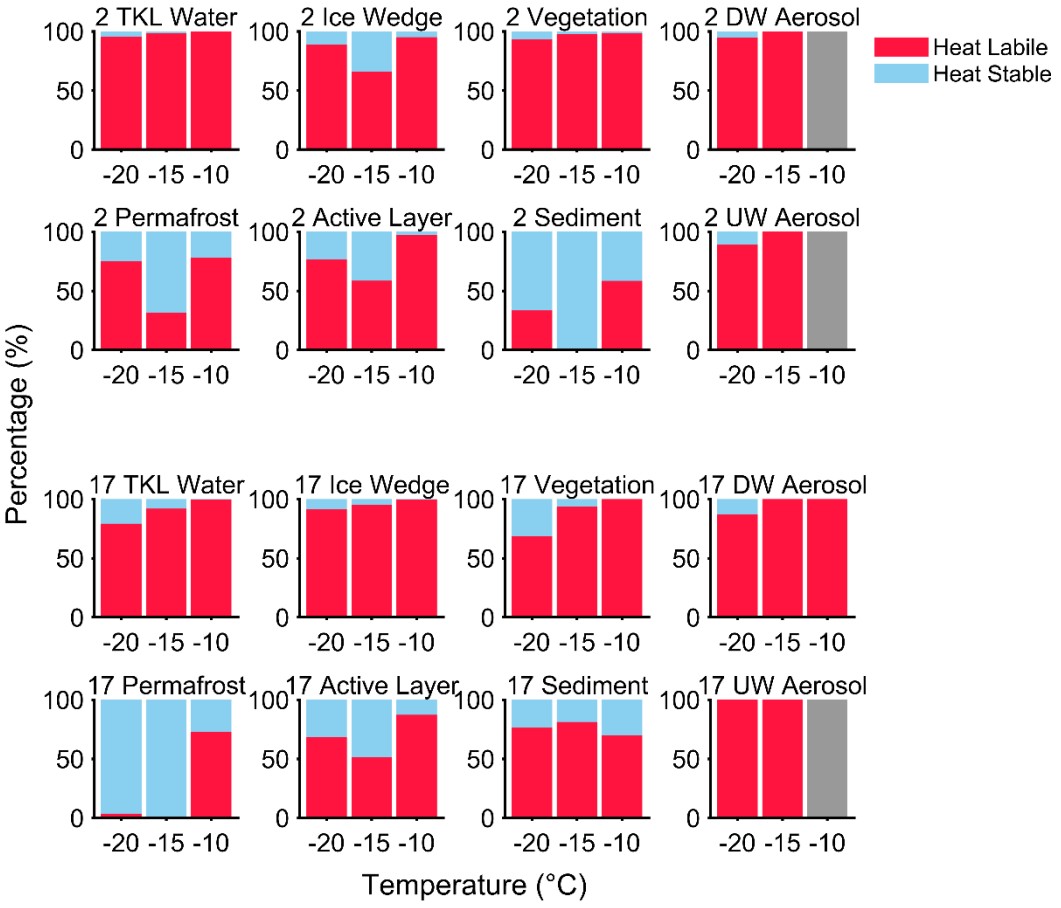

**Figure 8:** Histograms for samples collected on September 2nd (top) and September 17th (bottom), 2021, showing the percentage of INPs that are heat labile (sensitive to 95 °C heating: red) and heat stable (other: blue). Gray indicates INPs in the aerosol that were below the detection limit of the IS.

**3.3 Covariance of INPs with organic carbon in water**

Since water bodies can serve as a reservoir and source of atmospheric INPs, tracking their concentrations is important to better understand the Arctic INP budget. Water TOC concentrations have been used previously to normalize and derive relationships with INP concentrations in Arctic and North Atlantic Ocean sea surface microlayer samples (Wilson et al., 2015) and were found to overpredict corresponding North Atlantic INP aerosol concentrations (McCluskey et al., 2018). Based on heating to 95 °C (Fig. 8), the INPs in the water were predominantly heat labile (presumably organic), and therefore might correlate with TOC. Figure 9 confirms this hypothesis, with higher levels of TOC generally associated with increased INP

concentrations. Taken together, there is a strong correlation ($R^2$=0.85). However, when water types are treated separately, the correlations are weaker for the lagoon ($R^2$=0.12) and ocean ($R^2$=0.45), likely due to increased homogeneity. Therefore, we consider the complete landscape to cover the most variability, although TKLs on their own have the same coefficient of determination ($R^2$=0.85). As wave breaking and bubble bursting are hypothesized to be the main mechanisms of release of INPs from thawing permafrost into the atmosphere (Barry et al., 2023), this relationship suggests a potential means of

representing not only thawing permafrost, but other mixed thermokarst sources of Arctic INPs using TOC as a proxy variable.

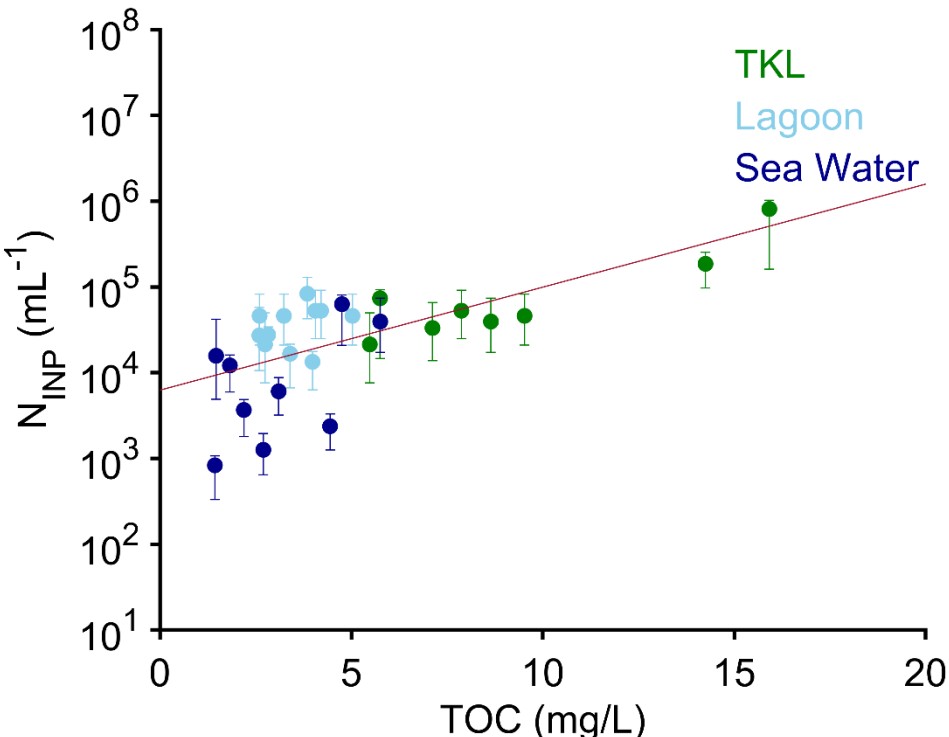

**Figure 9:** Water INP concentrations active at -15 °C (thermokarst lake water: green; lagoon: light blue; seawater: dark blue) versus total organic carbon (TOC). The red line gives an exponential best fit ($R^2$=0.85). 95% confidence intervals for the INP

concentrations are given.

### 4 Conclusions

    The Arctic is important to study sources of INPs due to limited previous observations in areas that have exhibited or are actively exhibiting thermokarst development and the sensitivity of mixed-phased clouds to INP concentrations. During

ARCSPIN, we comprehensively surveyed likely sources in an environment dominated by thermokarst processes, representing the first Arctic terrestrial-based source survey of INPs. Permafrost was found to be a large reservoir of INPs, with maximum

concentrations of $5\times10^8$ g$^{-1}$ at -15 °C, and furthermore, the highest concentrations were found closer to the coast, which could release more INPs with erosion. This analysis revealed many rich potential sources of INPs that are unaccounted for in current climate predictions. Water bodies have the potential to transfer INPs from these sources to the atmosphere, since they were

enhanced in the aerosol downwind of TKLs in all three cases measured, as well as the case with onshore winds off the ocean. However, a background level of INPs seem to exist given the relative insensitivity of the DOE INP concentrations to measured average wind direction and speed, and so the enhancement from TKLs and the ocean may be better viewed as periodic or a combined effect of passing over multiple TKLs.

This study represents the first attempt at INP source apportionment through PCA. Most source samples clustered

together within their groups and were separated on PC1 with overlap between the permafrost, sediment, and active layer, and additionally between water groups and vegetation. The permafrost had the most group variability, which may complicate future representation. Most of the aerosol INPs likely originated from a mixture of sources from separation on PC2 but spanning the range of source samples on PC1. The aerosol INPs were also found to be heat labile. These biogenic INPs could affect glaciation of Arctic clouds through their warm temperature activity. Heat tests on the potential source samples indicated high

heat labile populations from TKL water, vegetation, and ice wedge samples, but inconsistency and some temperature dependent insensitivity in the permafrost, sediment, and active layer samples. Differences between seemingly similar samples, such as permafrost and the active layer were also discovered through response to heat. The positive relationship found between INPs and TOC in the water suggests a potential approach to estimate INP concentration in models, similar to the approach suggested by McCluskey et al. (2018), since water can play a role for emission of INPs into the atmosphere. This connection

may be the most practical way to track current Arctic terrestrial INPs given the complexity of the landscape. To fully understand atmospheric Arctic INPs both now and in the future, knowing the permafrost coverage is critical due to it not only being a large reservoir of INPs, but also because it dictates the thermokarst landscape itself.

**Data availability**

DOI is TBD, will be published in the Arctic Data Center

**Supplement link**

TBD

**Author contribution**

JMC, SMK, PJD, and TCJH conceptualized the sampling campaign, and JMC, TCJH, KRB, and MNC participated in carrying

out the campaign (with guidance from TAD). KRB and TCJH processed the samples. KRB performed the sample analysis and wrote the manuscript with contributions from all coauthors.

**Competing interests**

The authors declare that they have no conflict of interest.

**Acknowledgements**

This work was supported by the National Science Foundation, Award No. 1946657. T. Douglas acknowledges the US Army Futures Command and the Assistant Secretary for the Army Acquisition, Logistics, and Technology Basic and Applied Research programs. Special thanks to Raelene Wentz, Cody Johnson, Matt Irinaga, Martin Edwardson, Harvard Brown, Jerry Brower, Eben Hopson, Ebony Brown, and Thomas Panningona (all from the Ukpeagvik Iñupiat Corporation) for the success of the ARCtic Study of Permafrost Ice Nucleation (ARCSPIN) campaign. Forest Banks and Rommel Zulueta from the National Ecological Observatory Network (NEON) Program/ Battelle are acknowledged for guidance and lending the SIPRE auger. Thank you to Amy Sullivan for use of her TOC instrument.

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
