# Peer review of "Active thermokarst regions contain rich sources of ice nucleating particles"

_EGUsphere, 2023_

## Author Comment (AC1)

**Response to Reviewer 1:**

Thank you to this anonymous reviewer that made helpful suggestions for strengthening this paper. As requested, we have performed further analyses, adding an additional sample to the heat labile INP section, as well as clarifications in the text and tables. We have made the changes and have hopefully addressed all concerns. Below, we provide a line by line response and detail all the changes (line numbers refer to the tracked changes version). Black text indicates the exact reviewer comment, green is our response, and blue is relevant text changes that appear in the manuscript.

Comments:

1.  Elaborate on how exactly INPs affect the surface energy budget via the Arctic Clouds

    The surface energy budget over the Arctic is impacted strongly by cloud properties, such as phase and particle size, and INPs affect these cloud properties. For example, liquid clouds over the Arctic strongly contribute to a positive cloud forcing, a warming effect, and so the level of INPs will change that balance.

    We now added a line:

    Lines 42-45: "They can alter the surface energy budget by impacting the cloud phase and optical thickness, as Arctic liquid clouds strongly contribute to a positive cloud forcing (Shupe & Intrieri, 2004). Replacement of ice with liquid in clouds has been shown to strengthen Arctic amplification, which is the enhanced regional warming due to phenomena such as the ice-albedo feedback (Tan & Storelvmo, 2019)."

    Shupe, M. D., & Intrieri, J. M. (2004). Cloud Radiative Forcing of the Arctic Surface: The Influence of Cloud Properties, Surface Albedo, and Solar Zenith Angle. Journal of Climate, 17(3), 616–628. https://doi.org/10.1175/1520-0442(2004)017<0616:CRFOTA>2.0.CO;2

    Tan, I., & Storelvmo, T. (2019). Evidence of Strong Contributions From Mixed-Phase Clouds to Arctic Climate Change. Geophysical Research Letters.

2.  The sampling height at some sites was 1.5m and for some 10m. Are both representative of the surface? How did you compare the two heights?

    This higher sampling height was chosen to get above roads and other buildings at the fixed site (photo below), in order to get an accurate picture of the INPs in the general Arctic boundary layer. For all of the field measurements, we were able to sample closer to the ground to get an idea of what the INPs were like from some of the potential periodic local sources (e.g., downwind of a thermokarst lake). Also, for transportability, having a 10 m sampling height in some of the remote sites would not be feasible. We believe the

measurements are both representative of the surface since they are in the boundary layer, as well as the averages for some types are similar: at -15 °C, the average INP concentration from thermokarst lakes was 0.02 L$^{-1}$ (1.5m), from the ocean was 0.04 L$^{-1}$, and from the DOE site (10m) was 0.01 L$^{-1}$. However, the INPs from the lagoon air (1.5m) were higher with an average of 0.16 L$^{-1}$.

[Figure]

3. In sample analysis section, please close the bracket for 20-fold dilutions (250 microL sample…

Thank you, Line 117 now reads "…(250 µL sample and 4750 µL 0.1-µm-filtered DI water)…".

4. Why is there data gap for 3,4, 12 and 16 September? The sampling period is very small!

The 4$^{th}$, 12$^{th}$, and 16$^{th}$ were days when only DOE measurements were collected. We added the dates of DOE sampling for completeness and to limit gaps (below). We were not able to sample in the tundra every day due to planning logistics, weather conditions, and availability of polar bear guards that accompanied us.

| Date (2021) | Name | Latitude (°) | Longitude (°) | Environment/ Collection type | Samples analyzed |
|---|---|---|---|---|---|
| 1-Sep | Emaiksoun Lake | 71.25057 | -156.77317 | Thermokarst lake (TKL) | DA, P, S, TW |
| 2-Sep | Untitled Lake 1 | 71.23529 | -156.30406 | TKL: upwind and downwind | DA, UA, AL, P, I, S, TW, V |
| 4-Sep | DOE | 71.32272 | -156.61506 | Fixed | A |
| 5-Sep | Point Barrow | 71.38535 | -156.46100 | Ocean and lagoon | A, LW, SW |
| 6-Sep | Nunavak Bay | 71.25240 | -156.87332 | TKL (brackish) | DA, AL, P, I, S, TW, V |
| 7-Sep | Elson Lagoon | 71.29581 | -156.26890 | Lagoon | A, LW |
| 8-Sep | Will Rogers/ Wiley Post Monument | 71.15311 | -157.06609 | Ocean: onshore | A, AL, P, S, TW, V |
| 9-Sep | Elson Lagoon DOE | 71.25556 71.32272 | -156.01580 -156.61506 | Lagoon Fixed | A, LW, SW A |
| 11-Sep | Untitled Lake 2 | 71.23806 | -156.60472 | TKL: upwind and downwind | DA, UA, AL, P, S, TW, V |
| 12-Sep | DOE | 71.32272 | -156.61506 | Fixed | A |
| 13-Sep | Mayoeak River | 71.25915 | -156.44528 | TKL (brackish) | DA, P, I, S, TW, V |
| 14-Sep | Will Rogers/ Wiley Post Monument | 71.15312 | -157.06609 | Ocean: offshore | A, AL, P, I, S |
| 15-Sep | Elson Lagoon DOE | 71.31522 71.32272 | -156.29845 -156.61506 | Lagoon Fixed | A, LW, SW A |
| 17-Sep | Emaiksoun Lake | 71.23097 | -156.77237 | TKL: upwind and downwind | DA, UA, AL, P, I, S, TW, V |
| 18-Sep | DOE | 71.32272 | -156.61506 | Fixed | A |

5. Make a table listing all the sources and their concentration of INPs

Thanks for the suggestion, we have now made a Table S1.

| Source | Mean INP Concentration (-15 °C) |
|---|---|
| Aerosol | $4.4*10^{-2}$ L$^{-1}$ |
| TKL | $1.2*10^{5}$ mL$^{-1}$ |
| Lagoon | $3.1*10^{4}$ mL$^{-1}$ |
| Seawater | $1.7*10^{4}$ mL$^{-1}$ |
| Active Layer | $3.6*10^{8}$ g$^{-1}$ |
| Permafrost | $1.1*10^{8}$ g$^{-1}$ |
| Sediment | $3.2*10^{7}$ g$^{-1}$ |
| Vegetation | $2.0*10^{6}$ g$^{-1}$ |
| Ice Wedge | $1.7*10^{5}$ g$^{-1}$ |

6. To divide INPs into heat labile and stable fractions, why only one day data was used? What was the rationale behind this?

   To strengthen the general conclusions, we analyzed another day and now added that day to the text, with the updated figure shown below.

[Figure]

7. Figure 7: choose different colour scale

   We now updated the color scheme, shown below.

[Figure]

8. Why is correlation weaker with increased homogeneity? Please elaborate.

We believe that the correlation is weaker due to both the INPs and TOC being similar and less variable from the relatively homogeneous lagoon, unlike the thermokarst lakes which cover a larger range due to being sometimes isolated and sometimes connected to the ocean via riverine transport, as well as varying depth and fetch. The ocean itself can have a large range with weather systems enabling waves to kick up sediment, while the lagoon is more sheltered.

9. INPs in the water are predominantly organic….are you referring to heat labile or stable organics?

We only tested the heat labile fraction thus far, so we can only say that. They are strongly heat-labile through sensitivity to 95 °C and so we will add that in the sentence for clarification.

Lines 321-323:

"Based on heating to 95 °C (Fig. 8), the INPs in the water were predominantly heat-labile (presumably organic), and therefore might correlate with TOC."

---

## Author Comment (AC2)

**Response to Reviewer 2:**

Thank you to Reviewer 2 for helpful suggestions to improve this manuscript. We have gone back and analyzed more TOC samples to include in the correlation with INPs plot. We have added several sentences to the conclusions and restructured the PCA and heat treatment section to focus on the results before a joint discussion. Additionally, we have throughout tried to ensure that we are not overstating and qualifying any results, while at the same time recognizing that this is an important start to better understanding of the INPs from a thermokarst region. We would prefer to leave the overall manuscript in a combined results and discussion format, due to its immediacy in addressing the findings, and as is allowed by the journal, and hope that we were able to address main points and changes in the response below. The line numbers refer to the number in the tracked changes manuscript. Thank you for your feedback and time. Black text indicates the exact reviewer comment, green is our response, and blue is relevant text changes that appear in the manuscript.

Review of "Active thermokarst regions contain rich sources of ice nucleating particles" submitted by Barry et al.

For this study, the authors collected a number of samples from different environments in the Alaskan Arctic close to Utquagvik for further analysis of the content of ice nucleating particles (INPs). Samples included air samples collected on filters up- and downwind of thermo-karst lakes (TKLs) but also at the shore and at a DOE site. Water samples were collected from e.g., the TKL themselves and from the ocean, and furthermore samples were collected from soils, sediments, the ice wedge and from vegetation.

Based on INP concentrations obtained from these samples, it was then argued that TKLs and the ocean provide a source for atmospheric INP.

On the other hand, a principle component analysis (PCA) was done, which I highly welcome. However, results from this analysis were only shortly described, and not involved in the overall interpretation of the results. Particularly, PCA seems to suggest that INPs from aerosol samples have diverse sources including particles from long range transport.

Also, heat treatment results were then introduced. Again, results from that seem to rather point to different INPs being present in the different sample types, with differences even between samples from permafrost and TKLs and all aerosol samples, and even differences between permafrost and active layer. This, however, again was not included in the main discussion and was not considered for the main conclusions of this study.

This manuscript can not be excepted in its present form. Instead, all data should be presented first, and then a joint interpretation of the data needs to be done. Given all the experimental evidence, to my understanding, the data do not support statements as e.g. the following from the abstract: "Arctic water bodies were shown to be important links of sources to the atmosphere in thermokarst regions."

After thorough revision, this study may become eligible for publication. However, currently I cannot support publication.

**Major comments:**

Lines 175-176: It is unclear to me how your data can provide this evidence. INP concentrations at DOE seem to be independent of airmass origin (and likewise of wind speed?). Passing TKLs does not add a lot of INP. For those examples where downwind and upwind measurements of TKLs were compared, wind speed was not discussed, although this is an important parameter for sea spray production.

Therefore, this conclusion seems to not be well informed and needs revision.

We added wind speed information to the TKL and DOE cases (Section 3.1). There was no obvious relationship between INP concentration enhancement and wind speed observed, as the largest TKL input was on the 17th, but that day did not have the highest wind speed. Previous studies have shown a power law relationship between primary marine aerosol and wind speed over the Southern Ocean (Moore et al., 2022; Sanchez et al., 2021) and North Atlantic (Saliba et al., 2019), and an exponential relationship between particle concentrations and wind speed over the Arctic Ocean (Leck et al., 2001). Freshwater studies are variable, showing a peak in 30 nm particles above approximately 3.5 ms$^{-1}$ (Slade et. al., 2010); however, there was relative insensitivity in a recent freshwater experiment between particle mass flux (PM$_1$) and wind speed, which was at least one order of magnitude lower from corresponding seawater experiments (Harb and Foroutan, 2022). The wind speeds experienced during this campaign were all typically less than 10 ms$^{-1}$ and would not necessarily correlate with INP production. Additionally, higher wind speeds may serve to dilute emissions. Aside from the wind, bubble bursting may occur through the release of methane in thermokarst lakes (ebullition: Walter et al., 2006), which could potentially serve as another mechanism to release INPs into the atmosphere. Altogether, we would not expect the INP concentrations to correlate with wind speed.

Lines 178-181: "Wind speeds were variable, with the averages on the 2nd, 11th, and 17th, of 9.3, 2.9, and 6.2 m s$^{-1}$ respectively, near and above the 3.5 m s$^{-1}$ threshold found by Slade et al. (2010) for freshwater aerosol enhancement. There was no obvious correlation between INP concentrations and wind speed, which agrees with recent relative insensitivity found between freshwater aerosol mass flux and wind speed (Harb & Foroutan, 2022)."

Lines 186-193: "This is despite slightly greater average wind speeds on the 14th (5.1 vs. 4.3 ms$^{-1}$). This analysis provides evidence for water bodies as potential vessels for transporting INPs to

the air under wind stress or alternative mechanisms such as methane bubbling up through TKLs, called ebullition (Walter et al., 2006). Primary marine aerosol has been found to have a power law relationship with wind speed over the Southern Ocean (Moore et al., 2022; Sanchez et al., 2021) and North Atlantic (Saliba et al., 2019), and an exponential relationship was found between particle concentration and wind speed over the Arctic Ocean (Leck et al., 2002). Enhancement in aerosol does not necessarily translate to enhancement of INPs. Further, wind speeds experienced during ARCSPIN were typically less than 10 m s$^{-1}$, with an average of 6 m s$^{-1}$ for the month of September."

Moore, K. A., Alexander, S. P., Humphries, R. S., Jensen, J., Protat, A., Reeves, J. M., Sanchez, K. J., Kreidenweis, S. M., & DeMott, P. J. (2022). Estimation of Sea Spray Aerosol Surface Area Over the Southern Ocean Using Scattering Measurements. Journal of Geophysical Research: Atmospheres, 127(22), e2022JD037009. https://doi.org/10.1029/2022JD037009

Sanchez, K. J., Roberts, G. C., Saliba, G., Russell, L. M., Twohy, C., Reeves, J. M., Humphries, R. S., Keywood, M. D., Ward, J. P., & McRobert, I. M. (2021). Measurement report: Cloud processes and the transport of biological emissions affect southern ocean particle and cloud condensation nuclei concentrations. Atmospheric Chemistry and Physics, 21(5), 3427–3446. https://doi.org/10.5194/acp-21-3427-2021

Saliba, G., Chen, C.-L., Lewis, S., Russell, L. M., Rivellini, L.-H., Lee, A. K. Y., Quinn, P. K., Bates, T. S., Haëntjens, N., Boss, E. S., Karp-Boss, L., Baetge, N., Carlson, C. A., & Behrenfeld, M. J. (2019). Factors driving the seasonal and hourly variability of sea-spray aerosol number in the North Atlantic. Proceedings of the National Academy of Sciences, 116(41), 20309–20314. https://doi.org/10.1073/pnas.1907574116

Slade, J. H., VanReken, T. M., Mwaniki, G. R., Bertman, S., Stirm, B., & Shepson, P. B. (2010). Aerosol production from the surface of the Great Lakes: GREAT LAKES AEROSOL PRODUCTION. Geophysical Research Letters, 37(18), n/a-n/a. https://doi.org/10.1029/2010GL043852

Harb, C., & Foroutan, H. (2022). Experimental development of a lake spray source function and its model implementation for Great Lakes surface emissions. Atmospheric Chemistry and Physics, 22(17), 11759–11779. https://doi.org/10.5194/acp-22-11759-2022

Walter, K. M., Zimov, S. A., Chanton, J. P., Verbyla, D., & Chapin, F. S. (2006). Methane bubbling from Siberian thaw lakes as a positive feedback to climate warming. Nature, 443(7107), 71–75. https://doi.org/10.1038/nature05040

Line 220: "have" should be "having". But the main point in this part of the text is, that it is unclear where this conclusion of "atmospheric importance" comes from. Just because there is some highly ice active material on vegetation, it does not mean that it is atmospherically relevant, as it is unclear how it would become airborne.

We removed the "thus have atmospheric importance".

Lines 238-241 now say: "Vegetation washings contained lower levels of INPs overall (average of $2 \times 10^6$ g$^{-1}$, SD=$1.7 \times 10^6$ g$^{-1}$ at -15 °C), but were the source of the warmest temperature INPs (Fig. S4), with a detected freezing onset as warm as -4 °C, likely indicative of populations of ice nucleation-active bacteria (e.g., Hill et al., 2014; Huang et al., 2021)."

Lines 250-251: Here it is very clear that the sequence in your text is not optimal. The results presented here make it necessary that earlier conclusions need to be revised. This should have been shown earlier, before presenting some of the interpretations and conclusions given above.

It would be good to first introduce all results, including those from heating and PCA. And only THEN start interpretations, in a new chapter, in which all results are discussed together! Your results are valuable, even if you cannot pinpoint the INP sources! But do not make statements that are later on contradicted by some of your own results.

We have restructured this section to present the results from the heating and PCA together before doing a joint discussion of them, acknowledging that they do not always provide the same answer as they are quite different in their approach to INP characterization. We have moved the TOC results and discussion to an entirely new section (3.3 Covariance of INPs with organic carbon in water).

Lines 251-282:

**"3.2 Source separation and characterization**

To better understand similarities and differences among thermokarst landscape sources and aerosol samples, we next present PCA and heat treatment results. PCA was applied to visualize how samples clustered within and between categories to relate INPs from sources to the aerosol. Figure 7 shows all source and aerosol samples colored by sample type. PC1 explains 47% of the variance while PC2 explains an additional 23% of the variance. Among sources, PC1 and PC2 separate water and vegetation (negative PC1 values) from permafrost and sediment samples (positive PC1 and negative PC2 values). The aerosol spans the range of collected samples along PC1, but is separated on PC2.

Samples can be further analyzed for characteristic INP differences through response to heating. In Figure 8, suspensions of all samples for September 2nd and 17th were heated to 95 °C to divide the INPs into heat labile and stable fractions. The heat labile fraction identifies putative biological INPs through protein denaturation, and the heat stable fraction may be organic or mineral. Among sources, the TKL water, ice wedge, and vegetation had the highest fractions of heat labile INPs, with above 95% at -10 °C, and above 90% at -15 °C except for the ice wedge on the 2nd (Fig. 8). The permafrost, active layer, and TKL sediment samples had high fractions of heat labile INPs at -10 °C (>70%: except for the sediment on the 2nd), while only the active layers and sediment sample from the 17th had greater than 50% heat labile INPs at both -15 and -20 °C.

The PCA revealed that while most sources, especially water and vegetation washings, cluster together within their groups, indicating relative homogeneity across location and time, there was substantial variability among permafrost. This suggests permafrost INP composition may be more heterogeneous, despite concentration comparibility to previous work presented in section 3.1. Both methods reveal differences between the permafrost and active layer, which were similar in INP concentrations (Table S1), but have relative PCA separation (Fig. 7) and heat sensitivity (Fig. 8), suggesting dissimilar INP populations. TKL sediments may also harbor distinct INP populations from other samples, with some uniqueness in heat labile fractions (Fig. 8), but similar values to permafrost on the PCA (Fig. 7). TKL sediments are comprised of former permafrost, but undergo loss of organic carbon, nitrogen, and phosphorous during the transition (Ren et al., 2022), which could have contributed to both measured similarities and differences to permafrost seen with the PCA and heat treatment. These tools should be seen as complimentary in providing pieces of evidence, as PCA utilizes the spectral shape of the original sample, while the heat treatment compares proportions in the two spectra.

All aerosol samples contained abundant heat labile INPs, with fractions of nearly 100% at all temperatures (Fig. 8), With PCA, the aerosol spans the range of collected samples along PC1, suggesting diverse sampled source types contributing to the collected airborne INPs (Fig. 7). As many source samples contained abundant heat labile INPs, with the most prominent being the TKL water, ice wedge, and vegetation, this analysis agrees with the PCA that multiple sources could have contributed. However, the PCA also uncovers that the aerosol are separated from most other samples along PC2, implying the aerosol additionally contained unsurveyed sources (e.g. other local or long range transport)."

Lines 254-255: It is not clear how the presence of heat labile INPs across the examined temperature spectrum supports the PCA conclusion of multiple INP sources. Check out e.g., Kunert et al. (2019), where for a single type of fungal spores heating affected the INP concentration across the whole temperature spectrum.

We changed the sentence as part of the restructuring of this section. Since the INPs are primarily heat labile in the aerosol, and since many of the sources have high fractions of heat labile INPs (temperature dependent), multiple sources may be contributing to what's in the aerosol, which is consistent with the PCA. None of the aerosol cumulative spectra are particularly steep (Figure S1) so it's unlikely that one INP population like a fungal spore dominated the aerosol (also the temperatures measured are much colder than the mean freezing temperatures reported in Kunert et al., 2019).

Lines 277-281: "All aerosol samples contained abundant heat labile INPs, with fractions of nearly 100% across temperatures (Figure 8), With PCA, the aerosol spans the range of collected samples along PC1, suggesting diverse sampled source types contributing to the collected airborne INPs (Figure 7). As many source samples contained abundant heat labile INPs, with the most prominent being the TKL water, ice wedge, and vegetation, this analysis agrees with the PCA that multiple sources could have contributed."

Line 262 ff, including Figure 9: Water TOC concentrations were not introduced in your study so far, so that part came as a surprise. Introducing your results first would have been good.

However, it is somewhat alarming that the correlation between INP and TOC concentrations only seem to work when you use all data from different water sources together. This also includes the one highest point in Figure 9, which certainly has a large influence on your resulting correlation. This seems to be a much too weak base to then suggesting the use of TOC concentrations as a proxy, particularly as you clearly said above, that it has been shown in literature that this does not work.

The TOC was introduced in the methods and is then introduced again in this section. In the revised manuscript, we performed TOC analyses on 10 additional water samples. This has added an additional TKL point at higher TOC concentrations. By including the varied water sources, we are able to cover a broader range of TOC values, and we find it notable that, despite differences, there appears to be a relationship evident in the data. If we could have sampled the lagoon or ocean at different times of year to cover greater variability within an individual water source, separate TOC correlations could be explored. The figure and our analysis propose a first estimate for linking INPs with a multitude of potential sources in a thermokarst environment but can certainly be improved and refined in the future.

Lines 269-270: Just a remark: It seems that some of your conclusions go back to the hypothesis from your earlier publication (that wave breaking and bubble bursting are main mechanisms for INP realease into the atmosphere). However, this is only a hypothesis! Be careful to discriminate between what you know and what you suggest, but also base that latter on the data you have.

We do say that it is a hypothesis: "As wave breaking and bubble bursting are hypothesized to be the main mechanisms of release of INPs from thawing permafrost into the atmosphere (Barry et al., 2023)...", but added the word "potential" as a qualifier.

Lines 328-331: "As wave breaking and bubble bursting are hypothesized to be the main mechanisms of release of INPs from thawing permafrost into the atmosphere (Barry et al., 2023), this relationship suggests a potential means of representing not only thawing permafrost, but other mixed thermokarst sources of Arctic INPs using TOC as a proxy variable."

Conclusions: Summarizing, the conclusions section gives statements which are not well rooted in the presented data. Much of it needs to be rewritten. This concerns specifically the following sentences:

Lines 284-285: There would only be rich potential for INP sources if you clarify how these INP may get airborne, by e.g., showing a relationship between INP concentrations and important parameters for sea spray generation via wind speed.

We believe that saying "rich potential sources" is an accurate characterization given the high numbers that exist in relevant mixed-phase cloud temperatures. "Rich" refers to their high concentrations, while "potential sources" acknowledges that, at this time, our study has not been able to definitively link aerosol to sources or release mechanisms. The sources sampled in our

work have the possibility of contributing to the aerosol, potentially through water bodies as we suggest in the next sentence. To our knowledge, this is the first study to sample this variety of in situ sources in this region. In our revised manuscript, we have made clearer the similarities we noticed between the INPs in the source samples and those in the air, while acknowledging that these links are not yet definitive.

Lines 285-286: Without knowing the influence of wind speed for the TKLs, TKLs and the ocean gave quite distinct pictures: while an enhancement of atmospheric INP due to the ocean was described for the one SINGLE (!) oceanic case, the enhancement after crossing TKLs was much weaker. This may be due to a saturation effect, but then, maybe also not many INPs were emitted from TKLs. Also, the PCA and the heat lability seem to show that the INP in the air and the TKLs and ocean are different.

Therefore, your data does not show that clearly, hence this sentence needs revision.

We adjusted the sentence to include the "three" for the TKL and "the case" for the ocean, but the numbers did show a significant enhancement with the pairwise t-test. We also now reiterate the findings from the DOE site here, to acknowledge that from an atmospheric perspective, a single TKL will not contribute many more INPs, but airmasses passing over a landscape full of TKLs will experience a repeated, cumulative and more pronounced enrichment. It would be ideal to get more measurements in the future (both from fixed sites and more downwind and upwind direct comparisons), but we can only speak from the measurements taken. It's true that the water INPs are different from aerosol (and even among water groups), and we wouldn't necessarily expect them to be similar since the addition of INPs is variable and temperature dependent.

Lines 345-349: "Water bodies have the potential to transfer INPs from these sources to the atmosphere, since they were enhanced in the aerosol downwind of TKLs in all three cases measured, as well as the case with onshore winds off the ocean. However, a background level of INPs seem to exist given the relative insensitivity of the DOE INP concentrations to measured average wind direction and speed, and so the enhancement from TKLs and the ocean may be better viewed as periodic or a combined effect of passing over multiple TKLs."

We also added an additional two sentences on the PCA results in the conclusions section.

Lines 350-353: "Most source samples clustered together within their groups and were separated on PC1 with overlap between the permafrost, sediment, and active layer, and additionally between water groups and vegetation. The permafrost had the most group variability, which may complicate future representation."

Line 290: Concerning the relationship between TOC and INPs, see my comment above. This is weak, mainly rooted on throwing a bunch of data from different sources together and having one outlier. To my understanding, this is not strong enough to allow for this statement.

We restructured and qualified this statement, based upon the results we have (and adding additional data points).

Lines 358-361: "The positive relationship found between INPs and TOC in the water suggests a potential approach to estimate INP concentration in models, similar to the approach suggested by McCluskey et al. (2018), since water can play a role for emission of INPs into the atmosphere."

**Minor and editorial comments:**

Line 24: Better replace "earth" with "containing".

We are trying to convey that permafrost is frozen earth material, so we would prefer to leave this as is.

Line 44: The lake spray aerosol production should be highly variable in time and depends on the amount of INPs entering TKLs, the drainage of the TKLs and factors of aerolization. I suggest to not mention such a precise time of "over 3 weeks" here.

We changed to "persist for weeks".

Lines 49-51: "If the material enters TKLs, its persistence in the water and release in lake spray aerosol could persist for weeks, with ice nucleation activity on a surface area basis up to and exceeding that of mineral dust (Barry et al., 2023)."

Lines 55-56: Already Creamean et al. (2018) described a transition from low to high INP concentrations during an ongoing Arctic spring, and Wex et al. (2019) showed a seasonal dependence for annual samples from several Arctic stations, which then, more recently, was corroborated in Li et al. (2022) for spring and fall data from Svalbard. Giving some more background here would be good.

We added a statement to the end of the sentence, acknowledging the seasonality observed that you mention.

Lines 60-62: "Most recently, a year-long observation of INPs in the central Arctic revealed a seasonal dependence with the highest concentrations found in summer (Creamean et al., 2022), similar to trends observed in other Arctic work (e.g., Creamean et al., 2018; Wex et al., 2019)."

Line 59: It could be worthwhile adding, that the active layer is the top soil layer in permafrost which is thawing during the summer.

We added "top" to where we introduce the active layer.

Lines 32-35: "Other thermokarst landforms include retrogressive thaw slumps, slope failures often triggered by the flow of material from the top seasonally frozen and thawed active layer, and thermokarst troughs and pits, low-lying areas created when ice-rich permafrost or massive ice features like ice wedges degrade."

Line 73: Explain the acronym "ATV".

Lines 79-81 now say: "Sites were chosen based on accessibility with all-terrain vehicles (ATVs) as well as to maximize areal coverage and diversity of terrain and weather conditions (e.g., targeting onshore versus offshore winds)."

Lines 83-84: "… filters were collected between 2 and 4 hours after deployment." How can they be collected after deployment? Do you mean after precleaning?

In this case, we were referring to after setting up the filters in the field, but we just removed those words.

Lines 89-90 now say: "The sampling height was approximately 1.5 m, and filters were collected after sampling 2 to 4 hours, depending on site."

Line 106: As ice wedges consist mostly of water, I wonder if it makes sense to treat as you did, i.e., treating them similar to the sediment, active layer and permafrost samples by giving their INP concentrations later on "per g of material"? Or might it be more useful to give them as "per L of water"?

Since the ice wedges were cored in the same way as the permafrost and can include bits of material like soil, the suspensions were prepared by weighing out 2 grams of material. Therefore, we treated them as per g of material. In reality, they are a hybrid between terrestrial and aquatic, and assuming a density of 1 g mL$^{-1}$, is equivalent to saying per mL of ice wedge water. For consistency, we left it as is.

Line 170: With "temperature dependent", do you mean the freezing temperature, and not the temperature of the surroundings!? Clarify!

Lines 181-182 now say: "We conclude that TKLs can generate INPs, but their impact on ice nucleation activity may be freezing temperature-dependent."

Line 211: Please write "Alaska" instead of "AK", as not everyone might understand the acronym immediately.

Lines 231-233 now say: "The permafrost INP spectra largely agree in concentration with permafrost samples from Fairbanks, Alaska, in Creamean et al. (2020) and with glacial outwash sediments from Svalbard, Norway (Tobo et al., 2019), with values between $10^8$ and $10^9$ g$^{-1}$ at -15 °C."

Lines 221-223: ("Ice wedges (Figs. S6 and S7) had values comparable to TKLs")

In your study, you related ice wedge and permafrost samples to the mass of collected material used for the INP measurements. On the other hand, INP concentrations from TKLs were related to the volume of collected water. Therefore: How can you then compare INP concentrations from TKLs to those from ice wedges? And how are they similar? Also, if what you wrote here were correct, the heat lability of the samples should be similar, which it only is to a certain extent. This needs revision.

Ice wedges and TKL are similar in the fact of the snow melt contributing to both, but they do have some differences in terms of heat labile INPs, as you mention, and ice wedges being frozen. This statement was only comparing their concentrations at -15 °C and not making any statement further than the similarity of that. They can be compared assuming a density of 1 g mL$^{-1}$ and we will add that in the sentence.

Lines 241-243: "Active layer samples (Figs. S6 and S7) had similar ice nucleation activities to permafrost, with an average concentration of 3.6x10$^8$ g$^{-1}$ at -15 °C. Ice wedges (Figs. S6 and S7) had values comparable to TKLs (assuming a density of 1 g mL$^{-1}$), with an average of 1.7x10$^5$ g$^{-1}$ (SD=1.7x10$^5$ g$^{-1}$) at -15 °C."

Lines 233-234: Can you explain which parameters are included in PC1 and PC2?

That is explained in the methods, lines 142-150, and is based on the slope in 2 degree temperature intervals and the log of the ratio of the INP concentration in 2 degree intervals and the average INP concentration.

Figure 7: Zoom in a little - there is ample empty space in your plot. It may be enough to use -4 to 4 for the PC2-axis, and -2 to 2 for the PC1-axis. Also, add lines for the two "0" values.

Thank you for the helpful suggestions, we have made those changes.

[Figure]

Line 280: Is it really still true that the Arctic has only a limited amount of previous observations? Just observing current developments in the past years, my impression is that more INP measurements were done in the Arctic than anywhere else. Better revise this sentence.

We have expanded the sentence to specify the region in which these observations are limited within the Arctic.

Lines 339-341: "The Arctic is important to study sources of INPs due to limited previous observations in areas that have exhibited or are actively exhibiting thermokarst development and the sensitivity of mixed-phased clouds to INP concentrations."

Fig S1: Use colors to discriminate between the different samples, such that the reader can see which samples were e.g. taken downwind of TKLs, at the DOE etc .

Thank you, we updated it accordingly.

[Figure]

Fig. S4: A different choice of color would be good: tan and salmon are very close and not easy to distinguish. (Also in other figures where the same colors were used.)

We updated the relevant figures to change the permafrost to a gray color.

[Figure]

**Literature:**

Creamean, J. M., R. M. Kirpes, K. A. Pratt, N. J. Spada, M. Maahn, G. de Boer, R. C. Schnell, and S. China (2018), Marine and terrestrial influences on ice nucleating particles during continuous springtime measurements in an Arctic oilfield location, Atmos. Chem. Phys., 18, 18023–18042, doi:10.5194/acp-18-18023-2018.

Kunert, A. T., M. L. Pohlker, K. Tang, C. S. Krevert, C. Wieder, K. R. Speth, L. E. Hanson, C. E. Morris, D. G. Schmale, U. Poschl, and J. Frohlich-Nowoisky (2019), Macromolecular fungal ice nuclei in Fusarium: effects of physical and chemical processing, Biogeosciences, 16(23), 4647-4659, doi:10.5194/bg-16-4647-2019.

Li, G. Y., J. Wieder, J. T. Pasquier, J. Henneberger, and Z. A. Kanji (2022), Predicting atmospheric background number concentration of ice-nucleating particles in the Arctic, Atmos. Chem. Phys., 22(21), 14441-14454, doi:10.5194/acp-22-14441-2022.

Wex, H., L. Huang, W. Zhang, H. Hung, R. Traversi, S. Becagli, R. J. Sheesley, C. E. Moffett, T. E. Barrett, R. Bossi, H. Skov, A. Hünerbein, J. Lubitz, M. Löffler, O. Linke, M. Hartmann, P. Herenz, and F. Stratmann (2019), Annual variability of ice nucleating particle concentrations at different Arctic locations, Atmos. Chem. Phys., 19, 5293–5311, doi:10.5194/acp-19-5293-2019.